# Effect of Different Culture Methods on Growth and Survival of the Snout Otter Clam, *Lutraria philippinarum*, in Bai Tu Long Bay, Vietnam

**Cao Truong Giang** [1,*], **Sarah Ugalde** [2,*], **Vu Van In** [1], **Tran Thi Thuy** [1], **Tran The Muu** [1], **Vu Thi Huyen** [1], **Dang Thi Lua** [1] , **Tran Thi Nguyet Minh** [1], **Trinh Dinh Khuyen** [3], **Le Van Khoi** [1] **and Vu Van Sang** [1,*]

[1] Research Institute for Aquaculture No. 1, Dinh Bang, Tu Son, Bac Ninh 16352, Vietnam
[2] Institute for Marine and Antarctic Studies, College of Science and Engineering, University of Tasmania, 20 Castray Esplanade, Battery Point, Tasmania 7004, Australia
[3] Aquaculture Farming Department, Faculty of Fisheries, Vietnam National University of Agriculture, Trau Quy, Gia Lam, Ha Noi 12406, Vietnam
* Correspondence: truonggiang@ria1.org (C.T.G.); sarah.ugalde@utas.edu.au (S.U.); vvsang@ria1.org (V.V.S.)

**Abstract:** This is the first study to examine the effect of three different cultivation methods (bottom-tray culture, suspended-tray culture, and beach/bottom culture) on the growth and survival rates of the snout otter clam, *Lutraria philippinarum*, after 12 months of grow-out cultivation from seed to commercial size. Analyses included weight, survival, shell size, and total fat. Although the results showed limited differences in growth among cultivation methods, survival rates were significantly different among three different culture methods. The bottom-tray cultivation method had the highest survival rate (76.5%), compared with suspended-tray cultivation (31.6%) and beach/bottom cultivation (52.5%). This demonstrates that the most suitable method for commercial snout otter clam farming is cultivation trays placed on the bottom of the substrate. Improving commercial farming of the species will support the development and expansion of aquaculture in Vietnam and elsewhere, while reducing the harvest pressure on wild populations.

**Keywords:** farming farm; fatness; growth; *Lutraria philippinarum*; survival rate

## 1. Introduction

Commercial farming of bivalves plays an important role in the economic and social development of many countries. In 2020, production of molluscs reached 17.7 million tons (USD 29.8 billion), which are mostly bivalves. Global exports of bivalve molluscs totaled USD 4.3 billion, representing around 2.8% of the value of global exports of aquatic products [1]. However, the global decline of nonmarine molluscs is causing increasing concern with many species now facing extinction. In particular, the otter clam, *Lutraria philippinarum* (Deshayes 1854) is commonly known as a bivalve that occurs in the Philippines, Thailand, China, Australia, and Vietnam. Moreover, it is one of the most commercially utilized species and has high demand in restaurants due to the delicious meat quality containing 11.63% protein, 0.42% carbohydrates, 1.22% minerals, and 18 different amino acids, some of which are essential amino acids [2,3].

In Vietnam, the snout otter clam is only confined to a very narrow range at the tidal flats along the limestone islands of Cat Ba, Ha Long, Bai Tu Long, and Co To, and even this resource has been seriously reduced in both density and biomass [4–6]. Moreover, the breeding of snout otter clams has been declining due to the quality of the breed and the uncontrolled rearing process, which has partly affected the survival rate of the farmed snout otter clams. In addition, the disease often occurs causing great damage to farmed snout otter clams. Therefore, this species is in danger of becoming threatened due to uncontrolled over-harvesting, which may eventually lead to population decline in Vietnam [3,7] and other countries [8].

According to many researches, molluscs have well-developed commercial cultivation methods that are popular since the methods are effective within the typical aquaculture constraints in many nations [9,10]. There are different types of cultivation methods applied depending on the species being cultivated, such as bottom culture where species are cultivated directly on the substrate, suspended culture where species are cultivated above the substrate using ropes, lines, trays, or rafts, and combination culture where more than one method is applied [11–13].

However, for the snout otter clams, the published literature were mainly on resources, biological characteristics, reproduction, and disease [4–6,8,14], with no current studies on the effectiveness of commercial cultivation methods. Therefore, this study aims to identify the most suitable and effective cultivation method for snout otter clam based on growth and survival at Bai Tu Long Bay, Northern Vietnam. The composition of phytoplankton in Bai Tu Long Bay is quite plentiful and diverse: Species composition of phytoplankton includes 210 species, 67 genera, 29 families, 9 orders, and 4 classes. Silicic algae account for 62% of total species with 130 species, 45 genera, 17 families, and 2 orders. Dinophyceae algae account for 37% of total species with 76 species, 20 genera, 10 families, and 5 orders. Heterokontophyta algae account for 1% of total species with 2 species, 1 genus, 1 family, and 1 order; Cyanophyte account for 1% of total species with 2 species, 1 genus, 1 family, and 1 order. The density of algae cells in the water is over 200,000 cells/liter on average and changed over the year [7]. Therefore, this area is suitable for the design of the experiment using different culture methods for snout otter clam. By improving cultivation methods, this will assist in developing and expanding the otter clam aquaculture industry not only in Vietnam, but also around the world by applying the most efficient culturing methods.

## 2. Materials and Methods

### 2.1. The Experimental Site

The experiment was conducted at a windtight bay in Bai Tu Long Bay, Northern Vietnam (20.971° S, 107.431° E). The site had a depth of 3 to 5 m and tidal range of about 3.5–4.0 m/day. This area had a typical all-day tidal regime, it experienced two high and two low tides of approximately equal size every lunar day (a semidiurnal tidal cycle), and a period of water intensity from 11 to 13 days. The salinity fluctuates according to two main seasons of the year: Dry season was from November to May with quite high and relatively stable salinity between 30.0 and 31.0‰, and the rainy season was from June to October with fluctuating salinity between 25.0 and 30.0‰. The average annual temperature was in the range of 22.6–24.0 °C, and the hottest time was between June and August (especially in July with 28.0 °C on average). The annual average surface water temperature was about 22.0–24.0 °C, and higher than the summer months (May to October, about 28 °C). In the winter months, temperatures are lower, and lowest in January with an average of about 17.8 °C. BOD5 content (COD, $N-NO_2^-$, $N-NO_3^-$, $N-NH_4^+$) were within the allowable range according to the Vietnamese standards for the quality of coastal water (TCVN 5943-1995) [3,7].

### 2.2. Experimental Design

The experiments were carried out from October 2008 to September 2009. The experiments utilized snout otter clams obtained from hatchery production at the Northern National Broodstock Center for Mariculture—Research Institute for Aquaculture No. 1, in Northern Vietnam. A total of 6750 snout otter clam seeds were from the selectively bred population. Initial shell length and height were 45.0 ± 1.5 mm and 16.0 ± 0.8 mm, respectively, and the whole body weight was 13.0 ± 0.7 g. The cultivation site of snout otter clam was located at Bai Tu Long Bay (20.971° S, 107.431° E) in Northern Vietnam.

### 2.3. Cultivation Methods

The snout otter clam was cultured and assessed by three different cultivation methods at Bai Tu Long Bay. The culturing density of spat for three methods was 30 individuals

per basket; therefore, each treatment/experiment used 750 individuals for assessment [2]. The cultivation methods included (1) Bottom-tray cultivation: Trays were placed on the sea bottom (trays had dimensions of 50 cm length × 40 cm hight × thickness of 25 cm, the bottom of the tray was lined with a net, then sand was poured with debris of mollusk shells 8 cm thick); (2) Suspended-tray cultivation: Trays were spread with sand 15.0–20.0 cm thick and hung under a bamboo raft 1.0–1.5 m from the water surface, and trays could rise and fall with the tides; (3) Beach/bottom cultivation method: Snout otter clams were driven into the bottom sand (culturing site is characterized by a shellfish-sand bottom) using a bamboo/wood stick to pierce the surface of the sand to create holes 6–7 cm apart (5 cm in depth) and one individual clam was placed into each hole [2]. Each of the three cultivations was replicated three times.

### 2.4. Sampling and Measurements

Sampling and measurements of snout otter clams in each cultivation method were conducted monthly from October 2008 to September 2009. Every month, 30 snout otter clams were sampled and processed to measure growth indicators. Whole snout otter clams were rinsed with water to remove sediments and weighed (whole body weight), checked for survival (survival), and the shell was measured (shell height and shell length). Then, the meat was removed using knives or trowels, weighed (*meat weight* and *shell* weight), and stored frozen for later analysis (total fat). All sampled snout otter clams were processed regardless of size. Stainless steel calipers with an accuracy of 0.01 mm and a digital scale with an accuracy of two decimal places were used.

### 2.5. Growth Biometrics

Biometrics were shell height (SH, the maximum distance between the hinge to the furthermost edge), shell length (SL, the maximum distance between the anterior and posterior margins), and whole body weight (BW, snout otter clams blotted dry with absorbent paper before weighing). Growth relationship of BW and SH (n = 6750; 30 individuals per tray, each tray was replicated three times) was estimated using the potential regression W = aLb; where W is the BW (g), L is the SH (mm), a is the intercept, and b is the slope.

Survival (%) was estimated as $(N_t/N_0) \times 100$, where $N_t$ is the number of live snout otter clams sampled, and $N_0$ is the number of snout otter clams at the beginning of the experiment. Monthly mortality ratio was calculated from $Z = (N_t - N_0)/N_0$, where $N_0$ is the initial number of individuals, and $N_t$ is the final number of individuals [15].

$$\text{Fatness (Meat Yield)\%} = [\text{Meat wet weight/total wet weight}] \times 100.$$

### 2.6. Statistical Analysis

Differences in mean SH, SL, and BW among the cultivation methods were analyzed with a one-way ANOVA using SPSS software (version 24.0) and GraphPad Prims 7 (https://www.graphpad.com, accessed on 5 May 2022). Values of each parameter averaged over the month were compared among the cultivation methods using Pearson correlation coefficient in SPSS software (version 24.0).

## 3. Results

### 3.1. Biometrics—Shell Length (SL) and Shell Height (SH)

After a culturing period of 12 months, from an initial average size of shell length of 45 mm, shell height of 16 mm, and body weight of 13 g at a stocking density of 30 individuals/m$^2$, to a period of 12 months with three experimental treatments, the shell growth (Figures 1 and 2) had no statistically significant difference ($p > 0.05$) between the bottom-tray and suspended-tray cultivation methods. They showed significant differences ($p < 0.05$) compared with beach/bottom cultivation methods ($p < 0.05$). Specifically, at the end of the experiments, the bottom-tray cultivation had higher growth rate than the two other cultivation methods in terms of shell length (75.46 ± 0.95 mm) and shell

height (24.34 ± 0.28 mm) in comparison with the suspended-tray cultivation (shell length: 73.15 ± 0.65 mm; shell height: 25.63 ± 0.29 mm).

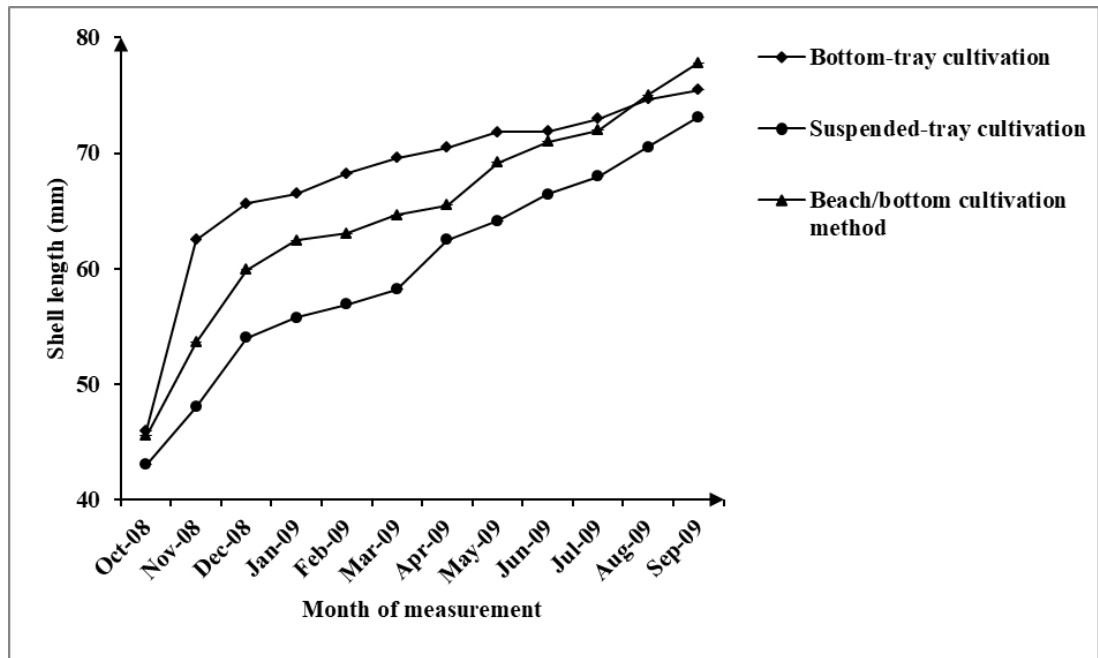

**Figure 1.** Experimet was conducted from October 2008 to September 2009, this results showed monthly average of shell length (SL) of snout otter clams in bottom-tray cultivation, suspended-tray cultivation, and beach/bottom cultivation methods. The values represented the average of three replicates ± SD ($p < 0.05$).

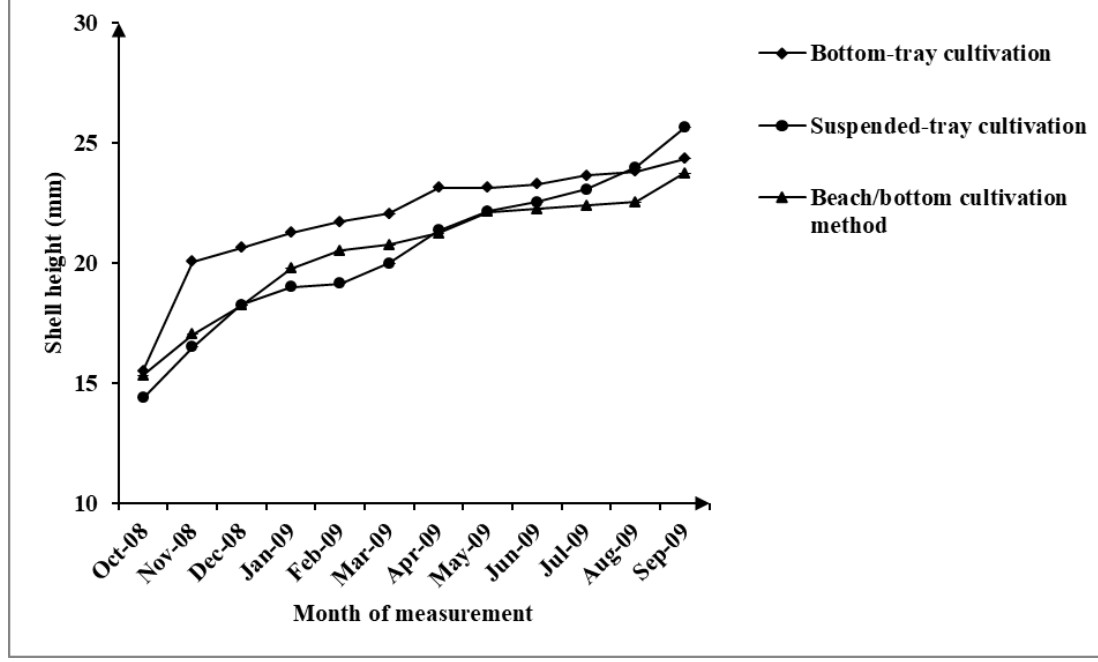

**Figure 2.** The experiment was conducted from October 2008 to September 2009, this results showed monthly average of shell height (SH) of snout otter clams in bottom-tray cultivation, suspended-tray cultivation, and beach/bottom cultivation methods. The values represented the average of three replicates ± SD ($p > 0.05$).

There was a strong linear relationship between SL and SH (Figure 3) with correlation coefficient $R^2$ = 0.910, with $R^2$ close to 1, the linear relationship was very strong. The $r^2$ value showed that 82.7% of the variation in the SL of snout otter clams was explained by the SH.

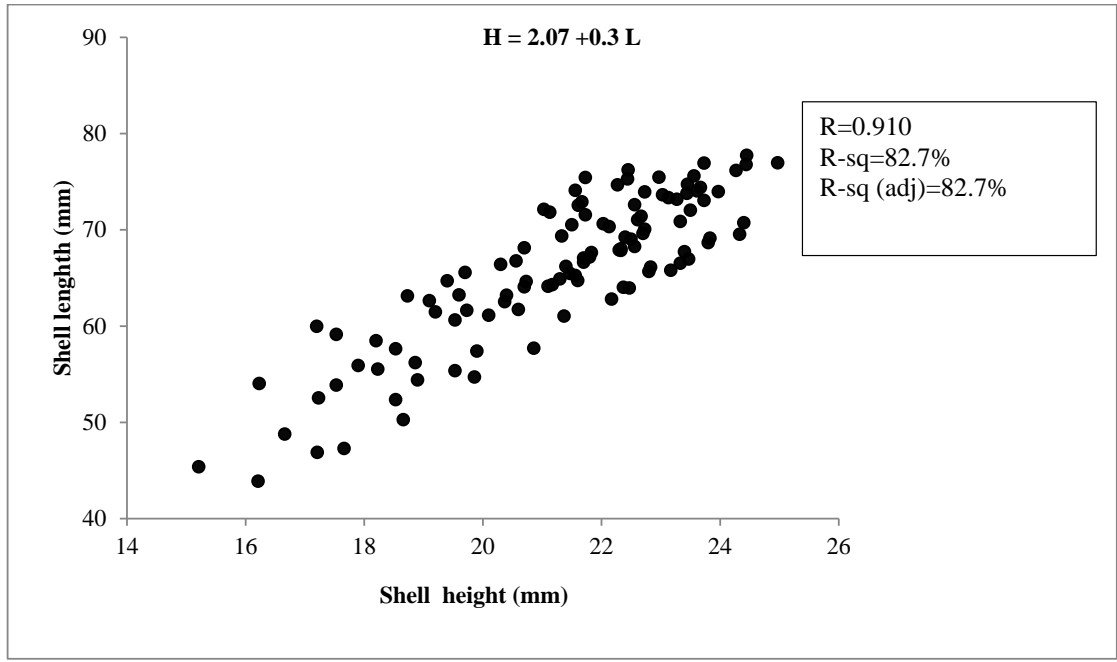

**Figure 3.** Morphological correlation for snout otter clams between shell length (SL) and shell height (SH), (all culturing methods combined).

### 3.2. Biometric—Body Weight (BW)

BW from the seed to the end of the experiment (i.e., commercial size) in all three cultivation methods (Figure 4) was not significantly different. Specifically, bottom-tray cultivation reached 54.28 ± 1.60 g (growth rate of 2.01 ± 0.35 g/month), suspended-tray cultivation reached 55.56 ± 0.41 g (growth rate of 3.64 ± 0.36 g/month), and beach/bottom cultivation reached 54.03 ± 1.55 g (growth rate of 3.25 ± 0.30 g/month). Correlation coefficient (r = 0.66) showed that when the otter clam grows one unit, the shell length also increases an equivalent unit (Figure 5).

### 3.3. Biometric—Fatness

The fatness of snout otter clams from the seed to the end of the experiments for all three cultivation methods (Table 1) did not show a statistically significant difference. The average fatness for bottom-tray cultivation reached 41.09 ± 0.83%, suspended-tray cultivation reached 40.32 ± 0.86%, and beach/bottom cultivation reached 42.80 ± 1.02% of fat.

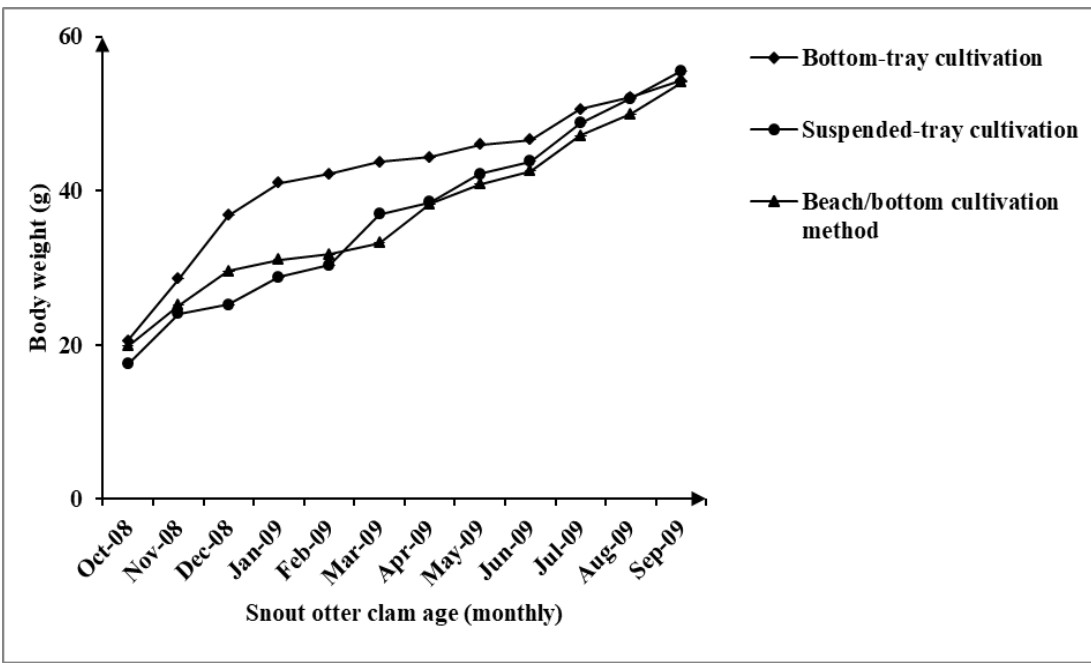

**Figure 4.** Study was conducted from October 2008 to September 2009, this results showed monthly average of body weight (BW) of snout otter clams in bottom-tray cultivation, suspended-tray cultivation, and beach/bottom cultivation methods. The values represented the average of three replicates ± SD.

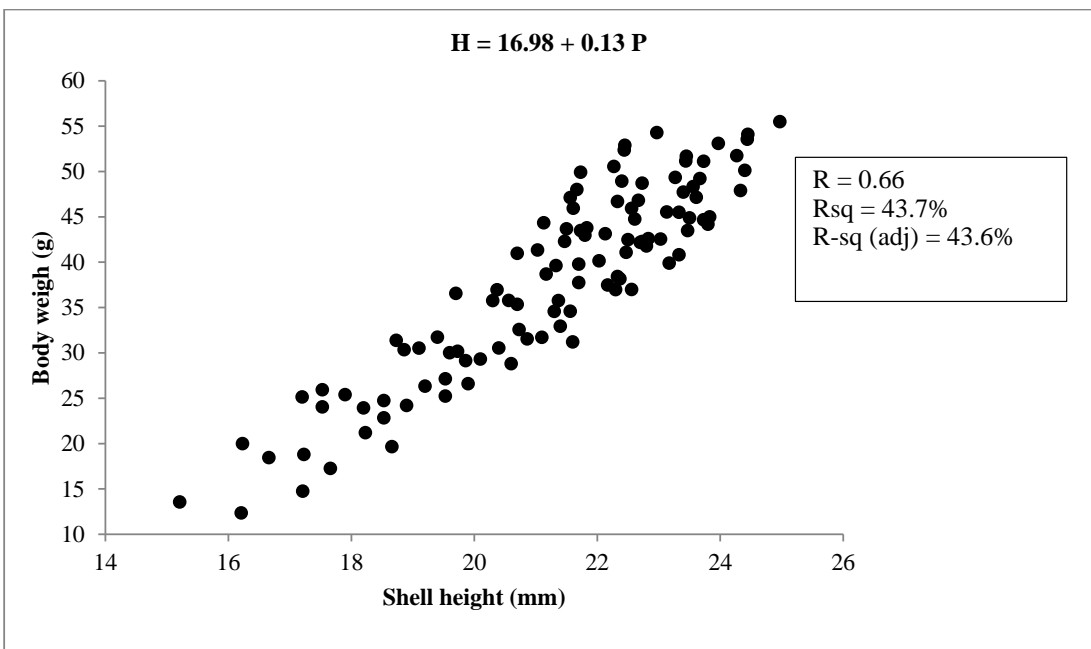

**Figure 5.** Morphological correlation for snout otter clams between body weight (BW) and shell height (SH), (all cultivation methods combined).

**Table 1.** Body weight, soft tissue weight, and fatness of snout otter clams in three different methods of growing.

| Time (Month) | Bottom-Tray Cultivation | | | Suspended-Tray Cultivation | | | Beach/Bottom Cultivation Method | | |
|---|---|---|---|---|---|---|---|---|---|
| | Whole Body Weight (g) | Soft Tissue Weight (g) | Total Fat (%) | Whole Body Weight (g) | Soft Tissue Weight (g) | Total Fat (%) | Whole Body Weight (g) | Soft Tissue Weight (g) | Total Fat (%) |
| October 2008 | 20.51 ± 0.93 [a] | 6.85 ± 0.99 [a] | 33.4 | 17.49 ± 0.77 [a] | 5.80 ± 0.27 [a] | 33.3 | 19.81 ± 0.81 [a] | 6.85 ± 0.22 [a] | 34.5 |
| November 2008 | 28.54 ± 1.02 [a] | 15.3 ± 1.54 [a] | 53.8 | 24.02 ± 0.91 [a] | 8.15 ± 0.41 [b] | 33.9 | 25.12 ± 0.79 [a] | 10.01 ± 0.30 [b] | 39.8 |
| December 2008 | 36.76 ± 0.98 [a] | 15.44 ± 0.60 [a] | 42.0 | 25.16 ± 0.69 [a] | 11.2 ± 1.18 [b] | 44.5 | 29.52 ± 0.82 [a] | 12.9 ± 0.25 [b] | 43.6 |
| January 2009 | 40.97 ± 1.32 [a] | 16.42 ± 1.15 [a] | 39.7 | 28.8 ± 0.64 [a] | 13.05 ± 0.47 [b] | 45.3 | 30.99 ± 1.06 [a] | 14.9 ± 0.35 [b] | 48.0 |
| February 2009 | 42.14 ± 0.85 [a] | 16.90 ± 0.86 [a] | 40.1 | 30.34 ± 0.98 [a] | 14.5 ± 0.50 [b] | 47.7 | 31.73 ± 1.47 [a] | 15.30 ± 0.34 [b] | 46.9 |
| March 2009 | 43.67 ± 1.16 [a] | 17.03 ± 0.92 [a] | 40.9 | 36.95 ± 0.93 [a] | 15.5 ± 0.28 [b] | 41.9 | 33.22 ± 1.41 [a] | 15.67 ± 0.42 [b] | 46.0 |
| April 2009 | 44.36 ± 1.15 [a] | 18.19 ± 0.89 [a] | 40.8 | 38.49 ± 0.84 [a] | 16.5 ± 0.64 [b] | 42.8 | 38.26 ± 1.00 [a] | 17.53 ± 0.30 [a] | 40.9 |
| May 2009 | 45.94 ± 1.03 [a] | 18.44 ± 1.19 [a] | 40.9 | 42.13 ± 0.79 [a] | 17.1 ± 0.77 [b] | 40.5 | 40.86 ± 0.82 [a] | 18.77 ± 0.38 [a,b] | 42.9 |
| June 2009 | 46.6 ± 1.69 [a] | 19.1 ± 1.38 [a] | 41.8 | 43.78 ± 0.73 [a] | 17.7 ± 0.71 [b] | 43.3 | 42.56 ± 1.19 [a] | 19.79 ± 0.34 [a] | 30.3 |
| July 2009 | 50.54 ± 1.61 [a] | 20.19 ± 0.02 [a] | 39.9 | 48.78 ± 0.66 [a] | 18.30 ± 0.47 [b] | 37.5 | 47.12 ± 1.67 [a] | 22.3 ± 0.23 [a] | 47.3 |
| August 2009 | 52.11 ± 1.47 [a] | 20.21 ± 0.03 [a] | 38.7 | 51.92 ± 0.52 [a] | 18.98 ± 0.56 [b] | 36.5 | 49.91 ± 1.17 [a] | 23.45 ± 0.41 [a] | 46.8 |
| September 2009 | 54.28 ± 1.60 [a] | 22.17 ± 0.02 [a] | 40.8 | 55.56 ± 0.41 [a] | 20.12 ± 0.71 [b] | 36.2 | 54.03 ± 1.55 [a] | 24.8 ± 0.26 [a] | 45.0 |

The values represent the average of three replicates ± SD. Different lowercase letters in the same row indicate statistically significant differences among different cultivation methods for the same month ($p > 0.05$, n = 30).

*3.4. Survival Rate (%)*

The survival of snout otter clams significantly differed among the three culture methods ($p < 0.05$). Specifically, the bottom-tray cultivation had the highest survival rate (76.5%) when compared with the beach/bottom and suspended-tray cultivation reaching 52.5% and 31.6%, respectively (Figure 6).

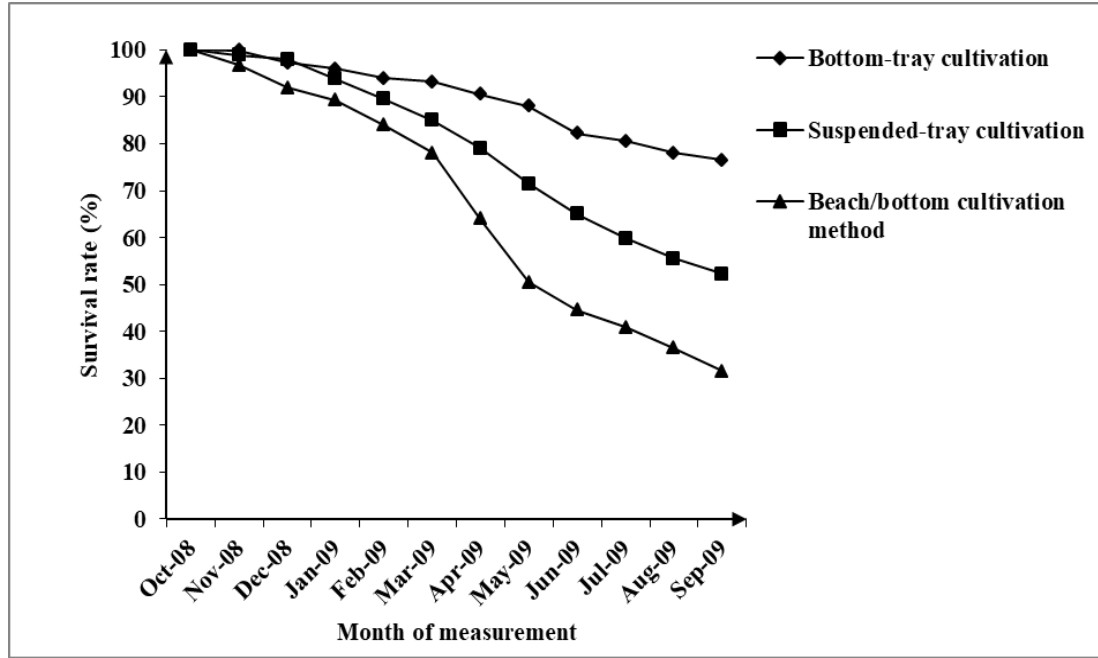

**Figure 6.** Study was conducted from October 2008 to September 2009, this results showed mean monthly survival rate (%) of snout otter clams over 12 months of cultivation in bottom-tray cultivation, suspended-tray cultivation, and beach/bottom cultivation methods.

## 4. Discussion

This study completed a growth assessment of cultivated snout otter clams with three cultivation methods. The growth assessment from seed to commercial size in all three cultivation methods showed statistical differences. In different parts of the Mediterranean, the differences in growth rates of bivalves are associated with food quality and quantity, temperature, upwelling intensity, sediment type, hydrodynamics, culture density, and fouling [16–19]. As already reported in other studies carried out with different shellfish under various experimental conditions, the cultivation method directly affects the growth and survival rates of each cultured species. Therefore, choosing a suitable commercial culture method is critical to ensure maximum yield. Benthic and suspended culture are the two most used methods of commercial culture of bivalves, and many farming models use these suitable methods efficiently to reduce costs. The experimental growth and survival results in this study showed that snout otter clams in different cultivation methods were well-developed from seed to commercial stage.

In addition, growth rates of bivalves are strongly influenced by fluctuations in the water environment [20]. Factors, such as temperature, salinity, nutrient content, flow rate, and water depth, influence the growth and survival rates of bivalve molluscs, especially when a limited amount of feed is available in the water column. In areas with temperate/subtropical climates, many studies have demonstrated that differences in the growth rates of bivalve molluscs are related to water depth [21–25]. Lodeiros, Rengel [22] conducted a study of scallops, *L. nodosus*, cultured at different depths and the results indicated that at a depth of 34 m, growth was low, the gonads did not appear to develop, and the survival rate was low. Lodeiros, Rengel [21] suggested that the low natural feed content was likely the main reason for the low survival rate of the scallops since the chlorophyll

*a* concentration at 34 m was 2.5 times lower at 21 m, and 6.8 times at 8 m in depth. Yu, Yang [26] reported that differences between two culture methods were mainly related to the quality of food available. Although the chlorophyll *a* concentrations were similar in the suspended culture and bottom culture, it is possible that the phytoplankton species composition differed between the surface and the bottom, and the light intensity, which was lower in deeper water, may have influenced the phytoplankton species and this could inhibit the growth of molluscs. In this study, a similar trend was observed for snout otter clams with the beach/bottom culture having similar growth rates to the suspended culture; however, the bottom culture has higher economic efficiency compared to the costly suspension model in terms of labor costs and effort [27].

However, in this study, the beach/bottom cultivation method had the lowest survival rate, while the bottom-tray cultivation method showed the highest survival rate. In this study, the differences in the results can be explained by variations in some factors, such as environmental conditions, the number of snout otter clams sampled, density, final BW, survival, and culture time. The bottom-tray cultivation method was slightly elevated off the seafloor, which may have also contributed to the improved performance over the beach/bottom cultivation method through the improved circulation and by allowing the excrement to fall more easily into the surrounding water Furthermore, the surface water layer is associated with a large amount of surface accumulation, and the changes in physical and chemical agents directly affect the ability to take in the food as well as the metabolism of the cultured species. Suspended matter that exists in the form of living microorganisms (epibionts) can adversely affect growth and survival rates due to competition for feed with cultured species [28] and reduced flow rate [29]. Therefore, the flow rate was the main factor affecting the growth of snout otter clam. If the flow rate was very high, it would limit the ability to food filtering ability [30]. According to Yu, Yang [26], fouling organisms on the lantern nets and scallop shells were present during their study; however, in this study, fouling was less in the bottom-tray cultivation method.

Temperature is also one of the main causes of high mortality [31]. Extremely hot summers, in combination with other factors, such as limited food avaibility and low water exchange, provide a stressful environment to the bivalves, resulting in mortalities. Reduced growth rates and high mortality typically occur in late summer until early autumn due to unfavorable natural conditions as well as physical and chemical factors (high temperature and low dissolved oxygen content), and thus affect the respiration and energy metabolism of the farmed species [32]. Therefore, the suspended cultivation method will be affected in summer due to the fact that the water temperature was higher near the surface than the bottom-tray cultivation method during summer, which may have led to the impact on growth [26]. In addition, the freezing winter temperatures can kill the bivalves if they are exposed to low water for a long period of time. Moreover, it can be the cause of the higher survival rate of bottom-tray cultivation and suspended-tray cultivation methods than the beach/bottom cultivation method since sea temperatures are usually higher than air temperature in severely cold conditions. Therefore, the bottom-tray cultivation method is recommended for fast development and high survival rate. Nevertheless, the reduced cost allows for a higher profitability over time. This was consistent with the research of Thanh (2012) [33], wherein the results of a survey on the current status of snout otter clam farming techniques in Quang Ninh, Vietnam showed that there were three farming methods, including direct stocking to the bottom (beach/bottom cultivation method) accounting for 13%, suspended-tray cultivation accounting for 14%, and bottom-tray cultivation accounting for 73%.

## 5. Conclusions

In conclusion, we conducted and selected the most suitable cultivation method for commercial aquaculture of snout otter clams based on the growth rate, survival, and fat content. Moreover, the study showed that the bottom-tray cultivation method had the highest survival rate, compared with the suspended-tray and beach/bottom methods.

Overall, these results strongly indicate that a better understanding of the cultivation of this species is needed. Furthermore, this first result may have important consequences for the optimization of commercial production that can assist in growing snout otter clams on a large scale.

**Author Contributions:** Development of the research concept, data collection and analysis, and writing original draft, edition of manuscript, C.T.G. and V.V.S.; research concept, data collection tools, and manuscript proofreading, S.U.; research concept and proofreading, V.V.I., T.T.T., T.T.M., V.T.H., D.T.L., T.T.N.M., T.D.K. and L.V.K. All authors have read and agreed to the published version of the manuscript.

**Funding:** Publishing was funded by the Australian Centre for International Agricultural Research (ACIAR) under the project 'Blue economy: Valuing the carbon sequestration potential in oyster aquaculture' (FIS/2020/175)". This research was funded by the Ministry of Science and Technology of Vietnam (MOST) to run the finishing technology of seed production and grow-out culture for snout otter clam (*Lutraria philippinarum*), grant number [KC06.DA16/06-10]. Snout otter clams were provided by the Northern National Broodstock Center for Mariculture (CATBANBC)—Research Institute for Aquaculture No. 1 (RIA1).

**Institutional Review Board Statement:** Not applicable.

**Informed Consent Statement:** Not applicable.

**Data Availability Statement:** Not applicable.

**Acknowledgments:** The authors would like to thank the Australian Centre for International Agricultural Research (ACIAR) via the project 'Blue economy: Valuing the carbon sequestration potential in oyster aquaculture' (FIS/2020/175)" for funding the open access publication fee of this paper. We advance our thanks to the Ministry of Science and Technology of Vietnam (MOST) who funded the research project. We are also grateful to the snout otter clam team at the CATBANBC, Ria1 for their technical assistance during the experimental periods, including data/sample collection and the management of the animals.

**Conflicts of Interest:** The authors declare that they have no known competing financial interest or personal relationships that could have appeared to influence the work reported in this paper.

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
