# Peer review of "Effect of Different Culture Methods on Growth and Survival of the Snout Otter Clam, Lutraria philippinarum, in Bai Tu Long Bay, Vietnam"

_2673-9496, doi:10.3390/aquacj3010005_

Round 1

Reviewer 1 Report

Dear Author(s),

In the present study, Cao et al. addressed The effectiveness of culturing methods on the growth and survival of the snout otter clam, Lutraria Philippinarum, in Bai Tu Long Bay, Vietnam. The manuscript presents an important and practical contribution to improving the growth and survival rate of snout otter clam.

- In general, the aim of the study is clearly defined. However, some important concerns need to arise from this study, particularly those regarding novelty and methodology.

- General revision of English is required

- Author should download the Article Form at the Aquaculture website and please correct it.

Introduction

- Line 49, insert in tables the same abbreviation you cite in the text.

- Line 61, please fix the typo ‘contries’

- Lines 61-62, please provide updated information

- Line 68, please correct the reference citation

- Table 1: replace ‘bivlaves’ by ‘bivalves’, ‘Locaions’ by ‘Locations’. Table 1 should be revised. It is quite difficult to read and understand. The scientific names of species are italicized.

- This study aimed to investigate different cultured methods for snout otter clam species, however, the literature information is careless and neglected. The relevant information should be mentioned. Each paragraph in the introduction section is not to link to each other.

Materials and Methods

- Section 2.1 The experimental site, grammar, and misprints revision is required. For example, Lines 82-83 – correct grammar.

- Quality of Figure 1 should be increased. 

- Lines 114-121, this information should move on to Introduction.

- In line 127, the authors stated that “a total of 6750 snout otter seeds were obtained. However, 300 out of 6750 were measured”. Please clarify.

- In section 2.4 Sampling and measurements, the authors state that “The meat was then removed using knives or trowels”, meaning that 30 snout otter clams were harvested monthly (according to the description from Lines 145-147 (???). If yes, the survival data are not reliable. Please confirm.

- Line 159. Please confirm the number of experimental animals.

- Did the authors check for normality of data? The description of significant letters should be explained (compared between rows or columns? or in each treatment?). Please clarify.

- Why standard error of the mean (SEM) was not taken in in this study? The SD has been taken but SD reflects variability within the sample while SEM may estimate the variability across the samples of a population.

- Please descript in detail for Figures and Tables Legends.

- In table 3, there is no significant difference in the body weight of experimental animals. Please remove all ‘a’ letters

- Figure 2,3, and 5 has a strange letter on the x-axis side. Please clarify.  

- The discussion is not appropriately written. In general, the authors need to discuss their results in a way that highlights the new knowledge they provide. Generally, all the discussion paragraph is required to be re-edit.

Conclusion: In my opinion, the authors must revise their manuscript. 

Author Response

Dear Reviewer,

Thank you so much for taking the time to give us comments. We have considered your comments carefully and made a thorough point by point revision as indicated below in the summary review report. On behalf of authors, I am attaching our revised form for your consideration.

Thank you and best regards,

Truong Giang Cao

Response to Reviewer 1 Comments

Point 1: Line 49, insert in tables the same abbreviation you cite in the text.

Response 1: Thank you very much for your comments. We have inserted the table 1.

Point 2: Line 61, please fix the typo ‘contries’.

Response 2: We have fixed countries.

Point 3: Lines 61-62, please provide updated information.

Response 3: Updated information were added from Line 75-97.

Point 4: Line 68, please correct the reference citation.

Response 4: The reference citation was changed in to number (Line 100).

Point 5: Table 1: replace ‘bivlaves’ by ‘bivalves’, ‘Locaions’ by ‘Locations’. Table 1 should be revised. It is quite difficult to read and understand. The scientific names of species are italicized.

Response 5: Table 1 was replaced ‘bivlaves’ by ‘bivalves’, ‘Locaions’ by ‘Locations’ and information in table 1 also edited for more clarity. The scientific names of species were italicized in table 1.

Point 6: This study aimed to investigate different cultured methods for snout otter clam species, however, the literature information is careless and neglected. The relevant information should be mentioned. Each paragraph in the introduction section is not to link to each other.

Response 6: We have added more information and made them link together.

Materials and Methods

Point 7: Section 2.1 The experimental site, grammar, and misprints revision is required. For example, Lines 82-83 – correct grammar.

Response 7: We have corected this section.

Point 8: Quality of Figure 1 should be increased. 

Response 8: The quality of Figure 1 was increased (Line 137-149).

Point 9: Lines 114-121, this information should move on to Introduction.

Response 9: This paragraph was moved on to the introduction section (Line 105-113).

Point 10: In line 127, the authors stated that “a total of 6750 snout otter seeds were obtained. However, 300 out of 6750 were measured”. Please clarify.

Response 10: 300 out of 6750 were removed (for further research) (Line 158).

Point 11: In section 2.4 Sampling and measurements, the authors state that “The meat was then removed using knives or trowels”, meaning that 30 snout otter clams were harvested monthly (according to the description from Lines 145-147 (???). If yes, the survival data are not reliable. Please confirm.

Response 11: The sampled oysters every month (Line 175-176) were not calculated for survival rate at the harvest. This method was successfully used in the other studies in oysters to estimate accurate survival (In et al., 2017; Vu et al., 2020).

Point 12: Line 159. Please confirm the number of experimental animals.

Response 12: The number of experimental animals was changed (Line 189).

Point 13: Did the authors check for normality of data? The description of significant letters should be explained (compared between rows or columns? or in each treatment?). Please clarify.

Response 13: The measurements of snout otter clam were kept when they were satisfied normality of data. We used the SPPS software version 24.0 to remove any data that did not follow normality of data. The description of significant letters in the rows showed the significantly considerable difference (P <0.05). 

Point 14: Why standard error of the mean (SEM) was not taken in in this study? The SD has been taken but SD reflects variability within the sample while SEM may estimate the variability across the samples of a population.

Response 14: Yes, thank you for your suggestions and comments. We would like to assess the variability of samples rather than expecting the whole population because the uniformity of snout otter clam is an important criteria in its industry as well.

Point 15: Please descript in detail for Figures and Tables Legends.

Response 15: We have added detail information for Figures and Tables Legends.

Point 16: In table 3, there is no significant difference in the body weight of experimental animals. Please remove all ‘a’ letters.

Response 16: Great, we removed the unnecessary letters “a” in the revised manuscript.

Point 17: Figure 2,3, and 5 has a strange letter on the x-axis side. Please clarify.  

Response 17: Thanks, we stands for months because there is no space in the figure 2,3 and 5 on the x-axis.

Point 18: The discussion is not appropriately written. In general, the authors need to discuss their results in a way that highlights the new knowledge they provide. Generally, all the discussion paragraph is required to be re-edit.

Response 18: Thank you for your grreat comments, we have added more information in the discussion section.

Point 19: Conclusion: In my opinion, the authors must revise their manuscript. 

Response 19: We have revised the whole manuscript. 

Reviewer 2 Report

The authors investigated the effectiveness of culturing methods on the growth and survival of the snout otter clam, lutraria philippinarum, in bai tu long bay, vietnam. They designed three methods of farming. This manuscript (MS) was clearly written and easy to understand. This work can help the sustainability of this species farming as little research in this area has been done. However, some minor issues significantly compromised the quality of this MS.

Minor comments

·       Line 29, please revise it; the sentence is not complete.

·       Line 83-84, revise how water and low water do not make sense.

·       Line 73, please make sure you defined the abbreviations for the first time in the MS.

·       Line 190-191, please revise it.

·       Table 3 and other parts, please delete the subset when there is no significant difference. Please check all tables and figures regarding this point.

·       Here and elsewhere, report P uppercase and italic (P<0.05).

·       Throughout the MS, if there is no significant difference, no need to report P-value.

·       Here and throughout the MS, please first mention the common name plus the scientific name, and for the rest of the MS, just report the common name.

·       Please update the introduction with recent works, as many studies are available from the last two years, which were not included in this section.

·       Please reorder the keywords alphabetically and capitalize each word.

·       I do not have any more comments and the author was written the next sections very well. There are some errors in the language that should be corrected.

Best regards

Author Response

Dear Reviewer,

We are grateful that you took the time out to leave us a review. We have considered your comments carefully and made a thorough point by point revision as indicated below in the summary review report. On behalf of authors, I am attaching our revised form for your consideration.

Thank you and best regards,

Truong Giang Cao

Response to Reviewer 2 Comments

Point 1: Line 29, please revise it; the sentence is not complete.

Response 1: Thank you very much for your suggestions, This sentence was rewritten.

Point 2:  Line 83-84, revise how water and low water do not make sense.

Response 2: We have rewritten this sentence (Line 122-125).

Point 3: Line 73, please make sure you defined the abbreviations for the first time in the MS.

Response 3: Yes, we have defined the abbreviations in the MS.

Point 4: Line 190-191, please revise it.

Response 4: we have revised and added information (Line 216-220).

Point 5: Table 3 and other parts, please delete the subset when there is no significant difference. Please check all tables and figures regarding this point.

Response 5: Yes, we removed all subsets in the table 3 where there is no significant difference.

Point 6: Here and elsewhere, report P uppercase and italic (P<0.05).

Response 6: Yes, we have added the P uppercase and italic for the whole revised manuscript. For example: Growth of body weight (g ± SD) over the months (P<0.05; SD = Standard deviation).

Point 7: Throughout the MS, if there is no significant difference, no need to report P-value.

Response 7: Yes, we removed all P-value in the places where there is no considerable difference

Point 8: Here and throughout the MS, please first mention the common name plus the scientific name, and for the rest of the MS, just report the common name.

Response 8: We have added the common name and plus the scientific name throughout the MS.

Point 9: Please update the introduction with recent works, as many studies are available from the last two years, which were not included in this section.

Response 9: We have updated information for the introduction.

Point 10: Please reorder the keywords alphabetically and capitalize each word.

Response 10: We have reodered the keywords alphabetically and capitalize each word (Line 39).

Point 11: I do not have any more comments and the author was written the next sections very well. There are some errors in the language that should be corrected.

Response 11: We have corected errors and edited language in the MS.

Reviewer 3 Report

Line 51: Need to be update to 2021  using FAO data.

Line 55: need to add a reference for this sentence.

Line 62: also need to be update to more recent data!

Line 62 to 65: add reference and example of farming.

Table 1: you can add source of seeds in south America that is from wild, artificial collectors and hatchery. Need to found a reference. You also have scallops in the North and south America from hatchery.

Line 132-142: need to present initial clams sizes (length and height) also explain the origem of the animals (hatchery, wild).

Line 165: need to check this equation. Fatness as described here is not appropriate. Do you want to present here condition index as described by Crosby and Gale (1990)? Please adapt your data!

Line 168: body weight is not appropriate data for daily growth rate, because can vary according to the reproductive cycle. For that is better to use height or length .

What was the temperature in the three cultivations conditions? What was the deep in the three culture sites ? and in the beach/bottom, animals were exposed to air? if yes for how long?

The three system were tested simultaneously? animals were all from the same batch?

Line 273-286: need to explain better the influence of farm conditions affecting survival and growth! The authors can compare with other species.

Conclusion should be presente based in your results!

Need English grammar editing.

Author Response

Dear Reviewer,

Thank you very much for your review. We have considered your comments carefully and made a thorough point by point revision as indicated below in the summary review report. On behalf of authors, I am attaching our revised form for your consideration.

Thank you and best regards,

Truong Giang Cao

Response to Reviewer 3 Comments

Point 1: Line 51: Need to be update to 2021 using FAO data.

Response 1: Thank you very much for your great comments. We have updated information using FAO data (Line 50-54).

Point 2: Line 55: need to add a reference for this sentence.

Response 2: We have added a reference for this sentence (Line 68, 70).

Point 3: Line 62: also need to be update to more recent data!

Response 3: Yes, we have updated more information (Line 75-90).

Point 4: Line 62 to 65: add reference and example of farming.

Response 4: We have added references and example of farming (Line 75-97).

Point 5: Table 1: you can add source of seeds in south America that is from wild, artificial collectors and hatchery. Need to found a reference. You also have scallops in the North and south America from hatchery.

Response 5: We have added source of seeds in south America in Table 1.

Point 6: Line 132-142: need to present initial clams sizes (length and height) also explain the origem of the animals (hatchery, wild).

Response 6: Snout otter clams obtained from hatchery production at the Northern National Broodstock Center for Mariculture - Research Institute for Aquaculture No. 1. Initial shell length and height was 45.0 ± 1.5 mm and 16.0 ± 0.8 mm, respectively, and whole body weight was 13.0 ± 0.7 g (section 2.2).

Point 7: Line 165: need to check this equation. Fatness as described here is not appropriate. Do you want to present here condition index as described by Crosby and Gale (1990)? Please adapt your data!

Response 7: Thank you very much for your comments. Fatness is a factor like the condition index, therefore we revised the equation/formula.

Point 8: Line 168: body weight is not appropriate data for daily growth rate, because can vary according to the reproductive cycle. For that is better to use height or length.

Response 8: Thank you for your great suggestions. Body weight may be affected by the reproductive cycle for calculating daily growth rate. We have removed equation/formula for daily growth rate.

Point 9: What was the temperature in the three cultivations conditions? What was the deep in the three culture sites ? and in the beach/bottom, animals were exposed to air? if yes for how long?

Response 9: The temperature in the three cultivations conditions was 27. The depth of three culture sites showed in Line 122 and the otter clams were raised in the rackets, therefore snout otter clams did not expose to the air.

Point 10: The three system were tested simultaneously? animals were all from the same batch?

Response 10: Yes, animals were tested simultaneously and the same batch to minimize the errors.

Point 11: Line 273-286: need to explain better the influence of farm conditions affecting survival and growth! The authors can compare with other species.

Response 11: We have explained the influence of farm conditions affecting survival and growth (Line 327-330).

Point 12: Conclusion should be presente based in your results!

Response 12: We have revised conclusion based on the results.

Point 13: Need English grammar editing.

Response 13: We have edited English grammar for whole MS.

Reviewer 4 Report

General comments:

This is an interesting paper that aims to compare the effects of different rearing mode on the growth performance and survival rate of a species of clam. The method is very simple, and the logic is direct. My major concern focuses on the statistical methods (see specific comments). I think that the present results may be not robust, and the results may have sharp changes after altering the statistical models, so I do not review the Discussion section in this review round. In addition, the language needs to be edited by a native speaker with biology background.

Specific comments:

L25: Please add the information for the background of this study.

L44: Please delete Taiwan.

L128-129: “45.0 ± 1.5 mm”, What does “1.5” indicate? SD or SE?

L133: I am confused to your experimental design. What number of animals in each treatment do you have? In line 146, you mentioned that 30 animals were sampled each month. But here you only mentioned 25 animals in each m2. Please elaborate this point.

L172: I think your statistical method may be improper. I mainly have two questions for this section. First, did you test the normality and homogeneity of your data before ANVA? Please note it in your paper. Second, it is ok for me that you used ANOVA to test weight, length and growth rate, but it is improper to use ANOVA to test survival rate and body fatness. The survival rate and body fatness are proportion data that have a range of [0, 1], so the robust method is generalized linear model with binomial or Beta error structure (depending on your data), rather than general linear model (i.e., ANOVA).

L192: Please add information on sampling size and type of error bar (i.e., SD or SE) to the Table caption. Add information for the implication of different small letters. The present form is very confused.

L195, 197, 216: Please add information on sampling size and type of error bar (i.e., SD or SE) to the Figure caption. In addition, add information for “SL” indication. Do not use abbreviation only.

L201, 220: Add the p value to the figure.

L204, 227: Please add information on sampling size to the Table caption. Add information for the implication of different small letters.

Author Response

Dear Reviewer,

We are so grateful for your review. We have considered your comments carefully and made a thorough point by point revision as indicated below in the summary review report. On behalf of authors, I am attaching our revised form for your consideration.

Thank you and best regards,

Truong Giang Cao

Response to Reviewer 4 Comments

General comments:

Point 1: The language needs to be edited by a native speaker with biology background.

Response 1: Thank you for your comments, the language in the MS was edited.

 Specific comments:

Point 2: L25: Please add the information for the background of this study.

Response 2: we have added information for the backgrount of the MS.

Point 3: L44: Please delete Taiwan.

Response 3: Taiwan was deleted (Line 56).

Point 4: L128-129: “45.0 ± 1.5 mm”, What does “1.5” indicate? SD or SE?

Response 4: As reported in the above comment, 1.5 is standard deviation. Therefore, we have added more information to clarify it in the revised manuscript: “45.0 ± 1.5 mm (mean + SD).

Point 5: L133: I am confused to your experimental design. What number of animals in each treatment do you have? In line 146, you mentioned that 30 animals were sampled each month. But here you only mentioned 25 animals in each m2. Please elaborate this point.

Response 5: Thank you very much for great detection. A total of 6750 animals were arranged in 3 experimental lots and each batch was repeated 3 times. Each single experiment used 25 trays and stocked 30 animals in the tray.

Point 6: L172: I think your statistical method may be improper. I mainly have two questions for this section. First, did you test the normality and homogeneity of your data before ANVA? Please note it in your paper. Second, it is ok for me that you used ANOVA to test weight, length and growth rate, but it is improper to use ANOVA to test survival rate and body fatness. The survival rate and body fatness are proportion data that have a range of [0, 1], so the robust method is generalized linear model with binomial or Beta error structure (depending on your data), rather than general linear model (i.e., ANOVA).

Response 6: Thank you very much for great detection. We have missed to add the statistical methods for the survival rate and body fatness. We have added more information to the revised manuscript in the methods (analyze the data). “The data were examined the normality and homogeneity before using the ANOVA methods in the SPSS software version 24.0. In addition, generalized linear model method with binomial written in ASReml famous software was used to analyze the survival rate and body fatness”. 

Point 7: L192: Please add information on sampling size and type of error bar (i.e., SD or SE) to the Table caption. Add information for the implication of different small letters. The present form is very confused.

Response 7: Yes, the sample size was 30 individuals per each single treatment and we used the standard deviation for the whole analysis. We have revised all unnecessary letters in the revised manuscript.

Point 8: L195, 197, 216: Please add information on sampling size and type of error bar (i.e., SD or SE) to the Figure caption. In addition, add information for “SL” indication. Do not use abbreviation only.

Response 8: We haved added the sampling size and type of error in the revised manuscript already. We also added information for “SL, SH and BW” indication in Figure 2, 3 and 5.

Point 9: L201, 220: Add the p value to the figure.

Response 9: Thank you very much for your suggestions. However, we have added the p-value in the caption of the figure.

Point 10: L204, 227: Please add information on sampling size to the Table caption. Add information for the implication of different small letters.

Response 10: Yes, we solved them in the above comments.

Round 2

Reviewer 1 Report

Dear Author(s),

The manuscript is much better than the previous version. However, some responses from the authors did not satisfy me.

1/ Author should download the Article Form at the Aquaculture website and please correct it. The link is provided as belove. The authors can select “Microsoft Word Template” or “LaTeX template” (https://susy.mdpi.com/user/manuscripts/upload/b25844cb6799b47018d10892a5a65870?form%5Bjournal_id%5D=335)

2/ Keywords, please replace “.” with “,”.

3/ The Introduction is too long but the information is not valuable for this study.

4/ Figure 1 displayed no meaning. The place for cultivation can be mentioned in Materials and Methods. There is no scientific information in Figure 1.

5/ Figures 2, 3, and 5 show a strange letter. The authors responded that they could not remove it even though they spent a lot of time (a month). It means that the authors do not understand the function of the program or software that they used.

6/ There is a miss between the letters “A” and “a” in Table 2.

7/ The experimental design for this study is not appropriate.

8/ The discussion is the weakest part of the manuscript. It mostly compares the current results with results on other, unrelated, species. These comparisons are inappropriate.

Author Response

Dear Reviewer,
On behalf of authors, thank you so much for taking the time to give us comments. We have considered your comments carefully and made a thorough point by point revision as indicated in the summary review report. 
I am attaching our revised form for your consideration.

Thank you and best regards,
Truong Giang Cao

Reviewer 4 Report

Lines 197-202, 251-255, 263-268: Did you truly revise your statistical methods and related results?

Author Response

Dear Reviewer,
On behalf of authors, thank you so much for taking the time to give us comments. We have considered your comments carefully and made a thorough revision as indicated in the summary review report. 
I am attaching our revised form for your consideration.

Thank you and best regards,
Truong Giang Cao

Round 3

Reviewer 1 Report

Dear author(s),

I would like to thank you for your responses.

I still have some comments and suggestions on this version as follows:

1/ The Template or Form has been provided in Aquaculture Journal, why do the authors not download and use it? The authors made the form by themselves, take a screenshot of the logo, and pasted it. I am really curious why? I also provided the link for downloading, please see previous comments.

2/ I also mentioned the references, it started from [8] (see Line 48). The authors have to correct it for their manuscript.

3/ Lines 83-86. The objectives of this study are not clear and specific.

4/ There are many errors typos or format in this version, the authors should check before submission. For example, Line 90 (experimentwas), Line 91 (3 to 5m), Line 124 (5cm), …

5/Figures 1, 2, and 4 still have Chinese Letters.

6/ Line 221, the author state that “…differed between the three treatments of culturing methods (ANOVA, P<0.05)”. Please describe it in a scientific way.

7/ Line 303, please correct the reference (which was written in Vietnamese). 

Author Response

I have considered and made a minor revision for our manuscript. I am looking forward to hearing from you for publication.

Sincerely
Dr Sang V. Vu

https://scholar.google.com/citations?user=3O4wnSsAAAAJ&hl=en

Response to Reviewer’s Comments

Point 1: The Template or Form has been provided in Aquaculture Journal, why do the authors not download and use it? The authors made the form by themselves, take a screenshot of the logo, and pasted it. I am really curious why? I also provided the link for downloading, please see previous comments.

Response 1: Thank you very much. I have updated it.

Point 2: I also mentioned the references, it started from [8] (see Line 48). The authors have to correct it for their manuscript.

Response 2: I have revised them.

Point 3: Lines 83-86. The objectives of this study are not clear and specific

Response 3: I have added more information to clarify this statement: Line 83-86: “it will help develop and expand otter clam aquaculture industry not only in Vietnam but also around the world via applying for the most efficient culturing methods”.

Point 4: There are many errors typos or format in this version, the authors should check before submission. For example, Line 90 (experimentwas), Line 91 (3 to 5m), Line 124 (5cm),

Response 4: I have gone through the manuscript and edit all errors typos.

Point 5: Figures 1, 2, and 4 still have Chinese Letters.

Response 5: I have corrected/editted all the errors and Chinese letters.

Point 6: Line 221, the author state that “…differed between the three treatments of culturing methods (ANOVA, P<0.05)”. Please describe it in a scientific way.

Response 6: I have revised: “The survival of snout otter clams significantly differed among the three culturing methods (P<0.05). Specifically, the bottom-tray cultivation had the highest survival rate (76.5%) when compared with the beach/bottom and suspended-tray cultivation reaching 52.5% and 31.6%, respectively”.

Point 7: Line 303, please correct the reference (which was written in Vietnamese). 

Response 7: I have corrected it.

Reviewer 4 Report

No further comments.

Author Response

(The authors gave the same response as above.)
